# DEALING WITH MISSING DATA USING ATTENTION AND LATENT SPACE REGULARIZATION

## ABSTRACT

Most practical data science problems encounter missing data. A wide variety of solutions exist, each with strengths and weaknesses that depend upon the missingness-generating process. Here we develop a theoretical framework for training and inference using only observed variables enabling modeling of incomplete datasets without imputation. Using an information and measure-theoretic argument we construct models with latent space representations that regularize against the potential bias introduced by missing data. The theoretical properties of this approach are demonstrated empirically using a synthetic dataset. The performance of this approach is tested on 11 benchmarking datasets with missingness and 18 datasets corrupted across three missingness patterns with comparison against a state-of-the-art model and industry-standard imputation. We show that our proposed method overcomes the weaknesses of imputation methods and outperforms the current state-of-the-art with statistical significance.

## 1 INTRODUCTION

Missing data is a common problem encountered by the data scientist. The consequences of missing data are often not straightforward and depend on the missingness generating process (Little and Rubin 2002). Choosing the best strategy to deal with missingness is critical when designing statistical or machine learning models that rely on incomplete datasets. A frequent complication is that the missingness generating process is often unknown, leading to assumptions about the missingness and potentially the introduction of bias into model training and inference (Davey and Dai 2020). The two most commonly employed strategies for dealing with missing data are to either drop data points where missing data exists or impute the values (Bertsimas et al. 2021). Both strategies can potentially introduce bias into a model if applied incorrectly (Little and Rubin 2002).

The 'impute and regress' strategy has been recently critiqued in the setting of machine learning predictors with the finding that an imputation method leads to a consistent prediction model if that model can almost surely undo the imputation, which questions the need for sophisticated imputation strategies (Bertsimas et al. 2018, 2021). The best case scenario for an imputation method is to replace missing unobserved variables with values that do not corrupt the model's predictive distribution from the true predictive distribution given only the observed variables. The worst case would be introducing a significant bias during training that increases the divergence of the model's predictive distribution from the true distribution. Indeed, Jeong et al. (2022) proved with an information-theoretic argument that there is no universally fair imputation method for different downstream learning tasks.

While practitioners continue to develop novel machine learning and statistical methods for imputation, relatively less effort has been made to create a framework to reason about models that can fit incomplete data with the notable exception of decision tree models (Gavankar and Sawarkar 2015; Jeong et al. 2022). In this paper, we consider the question of designing a model that trains and performs inference on only the observed variables, without imputation or data deletion.

### 1.1 RELATED WORK

Most recent work has focused on developing novel imputation techniques, such as auto-encoders, in order to build better imputations based on the structure of the data (Abiri et al. 2019). A consideration of learning and inference without imputation has been considered recently by Bertsimas et al. (2021)

who provided a theoretical approach to the deficiencies of imputation in predictive modeling, drawing an important distinction between statistical inference and prediction. They build on this approach to develop an adaptive hierarchical linear regression model that is capable of performing prediction in the presence of any missingness pattern. Similarly, Jeong et al. (2022) provide a method of inference without imputation using a decision tree approach, with a fairness-regularized loss function. Our work differs substantially from these approaches, whereby we show that training and inference using only the observed data is feasible using latent space representations and an entropy-based objective which ensures regularization against the potential bias of missingness.

## 1.2 Summary of contributions

1. Introduce a novel method of dealing with missingness by establishing a simple framework for reasoning about a model that fits and infers from the observed variables only.
2. Interpret the latent space representations in this framework with a measure- and information theoretic argument.
3. Empirically validate the theoretical properties of the latent space on a synthetic dataset with a latent space attention model.
4. Demonstrate the effectiveness of this model on benchmark datasets with corrupted data and real world datasets with missingness.

## 2 Interpreting missingness in a measurable space

We first describe a sample space $\Omega$ that is defined by an unknown data generating process and unknown missingness generating process. We then define three random variables:

$$X_d : \Omega \mapsto \mathbb{R}$$

$$Y : \Omega \mapsto \mathbb{R}$$

$$M_d : \Omega \mapsto \{0, 1\}$$

Where $d \in \{1, ..., D\}$ is the total number of possibly observed variables.

We can further specify, $X = \{X_1, ..., X_D\}$, which is the random variable $X : \Omega \mapsto \mathbb{R}^D$, and $M = \{M_1, ..., M_D\}$, which is the random variable $M : \Omega \mapsto \{0, 1\}^D$. A realization of $X$ is a vector representation of possibly observed variables and $Y$ is the outcome variable of interest. $M$ is a missingness vector where a value of 1 indicates a missing value and 0 an observed value.

If we consider the power set $\mathcal{U} = \mathcal{P}(X)$, then we can define a measurable space $(X, \mathcal{U})$. The impact of missingness results in a smaller $\sigma$-algebra such that $U = \{i : i \in \mathcal{U} \wedge j \in \mathcal{M} \wedge 1 \notin j\}$, where $\mathcal{M} = \mathcal{P}(M)$. Finally, we can define a probability space $(X, U, \mu)$ where a measure maps each combination of possible variables in $X$, defined in $U$, to a probability whereby $\mu : u \mapsto [0, 1]$, where $u \in U$.

An implication of our definition of the measure, $\mu$, is that where no missingness exists, each subset is equally likely, and the distribution is uniform. In the presence of a missingness generating process, some subsets are more likely than others and the distribution diverges from uniform.

How the probability distribution diverges from uniform can be considered with reference to the definitions of missingness originally described by Little and Rubin (2002). If $X_{obs} = \{x : x \in X \wedge m \in M \wedge m = 0\}$ and $X_{mis} = \{x : x \in X \wedge m \in M \wedge m = 1\}$ and we consider $M$ to be defined by some unknown parameter $\beta$, with a conditional distribution $f(M|X, \beta)$, then we can define three types of missingness:

**Definition 1.** *If $f(M|X, \beta) = f(M|\beta)$ then $M \perp X|\beta$ and data is considered to be missing completely at random (MCAR)*

**Definition 2.** *If $f(M|X, \beta) = f(M|X_{obs}, \beta)$ then $M \perp X_{mis}|X_{obs}, \beta$ and data is considered to be missing at random (MAR)*

**Definition 3.** *If $f(M|X, \beta) \neq f(M|\beta)$ and $f(M|X, \beta) \neq f(M|X_{obs}, \beta)$, then data is considered to be missing not at random (MNAR)*

We can extend these definitions to define conditional versions of $\mu$ with $\mu_{U|X}$ and $\mu_{U|X_{obs}}$. If $\mu(U) = \mu_{U|X}(U|X)$ then missing data can be considered MCAR. If $\mu_{U|X}(U|X) = \mu_{U|X_{obs}}(U|X_{obs})$ then missing data can be considered MAR. Finally, if $\mu(U) \neq \mu_{U|X}(U|X) \neq \mu_{U|X_{obs}}(U|X_{obs})$ then data is MNAR.

From this initial description of missingness, we can consider a naïve first attempt at predicting $Y$ from $X$ in the presence of missingness.

## 3 USING AN ENSEMBLE TO LEARN THE POWER SET OF FEATURES

As a starting point to developing our proposed method, we first consider a simple ensemble approach. We construct a hypothetical ensemble of $K$ models, parameterized by $\theta_k \in \Theta$, which performs the mapping for each combination of variables to our outcome space, $f_{\theta_k} : U_k \mapsto Y$, where $U_k \in U$, $k \in \{1, ...K\}$ and $K = |U|$. If the input vector $x \sim X$ contains missing values then only the complete subsets in $u \sim U$ with no missing values are used to update the models. To perform inference, we could simply choose the output of the function with the highest cardinality set. This strategy presupposes that all input data contains information related to the outcome, and feature selection has already occurred prior to model creation. In this setting, this model definition allows us to make a prediction, $\hat{y}$, given any subset of available variables from $X$.

This approach is only valid for datasets with missingness defined as MCAR as there are two implicit strategies occurring. The first is simply excluding the variable with missing data from our model by defining our subsets in $U$ which exclude that variable. This is always a valid strategy as it carries no risk of introducing bias into the remaining data, at the cost of losing information associated with that variable. The second implicit strategy is that in subsets that include variables with missing values, we remove all items in that subset that have a missing value. This is akin to dropping rows in a tabular dataset. By dropping missing data that is either MAR or MNAR, bias is introduced into the remaining data and the resultant model that is fitted to that data will incorporate that bias. In the "ensemble of $K$" models described above, models fitted on subsets with missing data will incorporate the biases of a dropping data strategy.

## 4 INCORPORATING MISSINGNESS IN A LATENT SPACE TO OVERCOME BIAS

In order to overcome the deficiencies of the previous formulation we could try to introduce a latent space representation for $U$ in our model and attempt to incorporate missingness into this latent space. We redefine $f_{\theta_k}$ to map each input vector in $U$ to a latent space $Z$, $f_{\theta_k} : U_k \mapsto Z$, which is shared across all possible subsets and define a second model, parameterized by $\Phi$ that maps Z to our output space, $g_\Phi : Z \mapsto Y$. We have now defined an "ensemble of compositions", $g_\Phi \circ f_{\theta_k}$.

Previous work by Boudiaf et al. (2020) has derived the result that when a latent space model is fit with a cross-entropy objective, $\mathcal{H}(Y; \hat{Y}|\hat{Z})$, this is equivalent to maximizing the mutual information $I(\hat{Z}; Y)$.

Using this result, we can reason about the representations of subsets in the latent space. Consider two subsets $U_i, U_j \in U$, which have identical variables, but one has an additional variable $|U_i - U_j| = 1$. If both $U_i$ and $U_j$ contain the same amount of information with respect to the outcome, then they will be clustered to the same location in latent space.

To formalize this clustering effect in terms of a measurable space, we can define a measure space for our dataset $(\mathcal{D}, \mathcal{A}, \mathcal{I})$, such that $\mathcal{D} = \{\{x, y\} : x \in X \wedge y \in Y\}$, $\mathcal{A} = \{\{u, y\} : u \in U \wedge y \in Y\}$, and $\mathcal{I}$ is a measure based on mutual information defined as $\mathcal{I} = I(f(U_k); Y)$, where $f$ is a general function that maps a subset, $U_k \in U$ to the latent space $Z$.

Mutual information is a correct basis for a measure in this measurable space as it satisfies the properties of non-negativity, mapping to zero for the empty set, and countable additivity across the subsets defined in $U$. The property of countable additivity relies on the independence of their latent space representations of $U_k$. This independence arises in two ways. Firstly, although we do not guarantee $X_i \perp X_j$ for $X_i, X_j \in X$ there is an independence across subsets of $X$ based on the construction of $U$, such that $U_i \perp U_j$ for $U_i, U_j \in U$. The independence of the subsets $U_k$ guarantees the independence of their latent space representations. Secondly, we can notice that $\mathcal{I}$ is

really measuring the mutual information of $Z|U_k$, and in the conditional universe of the latent space $Z|U_i \perp Z|U_j$.

Using our measure, $\mathcal{I}$, and model definition we can formally define the effect of the mutual information on the latent space.

**Proposition 1.** *If* $\mathcal{I}(U_i; Y) = \mathcal{I}(U_j; Y)$ *where* $i, j \in \{1, ..., K\}$ *and* $|U_i - U_j| = 1$, *then* $f(U_i) = f(U_j) = \hat{Z}|U_i = \hat{Z}|U_j$.

We can broaden this equality with a geometric interpretation.

**Proposition 2.** $|\mathcal{I}(U_i; Y) - \mathcal{I}(U_j; Y)| \propto d(f(U_i), f(U_j))$ *where* $i, j \in \{1, ..., K\}$, $|U_i - U_j| = 1$, *and* $d$ *is a distance metric in the latent space, such as the Euclidean distance.*

The effect of Proposition 2 is that the latent space representations for each $U_k$ in our "ensemble of compositions" cluster according to the mutual information. Additionally, there is a guarantee that the model learns representations for high cardinality subsets that are regularized to the highest information but lowest cardinality subset. This obviates the need for explicit feature selection, as non-informative features are not represented in the latent space.

**Remark 1.** *If* $Y$ *is a Bernoulli-distributed variable parameterized by the output of* $g$, $Y = Bern(g(f_k(U_k)))$, *then in the extreme case where no input features carry information with respect to our outcome all subsets will cluster to the latent space representation of the empty set, which is mapped to the point of highest entropy in* $Y$.

In order to understand how the latent space representation can help with missingness, we need only consider what happens to $\mathcal{I} = I(f(U_k); Y)$ as missingness affects $U_k$. If an unknown missingness generating process affects $U_k$, then $\mathcal{I} = I(f(U_k); Y)$ will decrease. This will be true whether the missingness is MCAR, MAR or MNAR. This reduction is directly represented in the latent space. When missingness is present in $U_k$, the representation in $Z$ is then regularized to the lowest cardinality and highest information subset.

Finally, it is interesting to note that there are many cases where missingness generating processes of the previously described patterns may also provide information related to the outcome (Li et al. 2018). This is commonly encountered, for example, in healthcare datasets where healthy people often have missing data because there was no indication to perform a test. We can define this property as arising when $I(M_d; Y) > 0$. In our latent space model, $\mathcal{I} = I(f(U_k); Y)$ for a subset $U_k$ which is affected by informative missingness may actually increase.

## 5 OUR APPROACH: USING ATTENTION TO MODEL THE LATENT SPACE

There is an obvious limitation to the ensemble approach, which is poor computational scaling as the number of models in the ensemble, $K$, increases exponentially with the dimensions of the dataset. Rather than defining a set of functions $f_{\theta_k}$ for each $k \in 1, ..., K$, we can define a general function $f_\theta$, which can map each $U_k$ to the latent space, as we did in Propositions 1 and 2. A natural choice for such a model that can take heterogeneous length inputs is to adapt an attention based model normally used to solve sequence tasks.

Scaled dot product attention has been used to construct state-of-the-art sequence models such as the Transformer (Vaswani et al. 2017). The Transformer maps an input sequence $(x_1, ..., x_d)$ to a latent space representation $(z_1, ..., z_d)$, from which a decoder outputs a sequence of $(y_1, ..., y_m)$. Here each $x, y, z \in \mathbb{R}^{embed}$, which is the embedding dimensionality. The similarity to the aforementioned "ensemble of compositions" can begin to be appreciated at this point. To approximate the ensemble with a Transformer architecture we must add two components. Firstly we add a feature specific embedding network that maps our input vector to some higher dimensional embedding space, $f_d : \mathbb{R} \mapsto \mathbb{R}^{embed}$. Then we apply a Transformer-style model where the output is instead $z \in \mathbb{R}^{embed}$, which is then mapped to the output space, $g_\phi : \mathbb{R}^{embed} \mapsto \mathbb{R}^{out}$. In this model, the output of the Transformer model $z \in \mathbb{R}^{embed}$, is analogous to the latent space in the previously compositional model and the function $g_\phi$ is serving the same purpose in both models.

It is important to note that while the compositional ensemble explicitly trains a separate model for each subset, the Transformer must train on randomly generated subsets at each training step. This

can be achieved using dropout to stochastically create a mask for the input features applied at each attention module. Concrete dropout has been shown to enable feature ranking and applying it at the feature level as a subset sampling process means that more informative features are included in the subsets more frequently (Chang et al. 2018). This choice of model design helps further regularize the output of the model toward the lowest cardinality, highest information subset.

Finally, we introduce our method as a latent space attention model (LSAM), which can deal with missingness 'out-of-the-box'. In summary, this approach uses a neural network to embed each feature, then maps the set of available embeddings to a latent space using a Transformer-style network. It is the latent space representation which is regularized against missingness as it trains. The latent space representation is then mapped to an outcome space using a neural network. The pseudocode for the training procedure is demonstrated in Algorithm 1 and training details can be found in the supplementary materials.

---

**Algorithm 1:** Latent space attention model training procedure. A stochastically generated mask, drawn from a concrete distribution at each training step, ensures the model learns from different subsets of variables.

---

Initialize network parameters $\rho, \psi, \theta, \phi$;

**for** *epoch in Epochs* **do**

    **for** $x, y$ *in batched($X \in \mathbb{R}^{n \times d}, Y \in \mathbb{R}^{n \times out}$)* **do**

        $mask \sim Concrete(\rho)$ ; /* Sampling feature mask for subsetting */

        $e \leftarrow f(\psi, x)$ ;        /* Embedding network $f_\psi : \mathbb{R}^d \rightarrow \mathbb{R}^{d \times embed}$ */

        $z \leftarrow t(\theta, e, mask)$ ;    /* Attention network $t_\theta : \mathbb{R}^{d \times embed} \rightarrow \mathbb{R}^{embed}$ */

        $\hat{y} \leftarrow g(\phi, z)$ ;      /* Output network $g_\phi : \mathbb{R}^{embed} \rightarrow \mathbb{R}^{out}$ */

        $g \leftarrow \nabla_{\rho, \psi, \theta, \phi} l(y, \hat{y})$ ;   /* Gradient of cross-entropy loss */

        $\rho, \psi, \theta, \phi \leftarrow update(g, \rho, \psi, \theta, \phi)$

    **end**

**end**

---

## 6 EXPERIMENTS

Models are built using JAX and trained using adaptive stochastic gradient optimization with early stopping (Kingma and Ba 2017; Bradbury et al. 2018; Zhuang et al. 2020).

Descriptions of hyperparameter optimization and model training can be found in the supplementary materials. Code for the model and experiments is available at https://anonymous.4open.science/r/Missingness-AF94/README.md.

### 6.1 MODEL CHARACTERISTICS

We first explore the properties of our proposed model with regard to the previously defined latent space using a toy dataset. Our aim is to show the effect of both the mutual information of a predictor on the latent space representations, and the effect of missingness on these representations.

#### 6.1.1 LATENT SPACE REPRESENTATION

In order to explore the representation of input variables in the latent space, we use the classic 2-dimensional synthetic spiral dataset. This is a non-linear binary classification problem with two input variables, $x1$ and $x2$, representing the x and y axes respectively. The 2 dimensional dataset is augmented with 2 additional variables. The first, $x3$, is random noise from a standard Gaussian carrying no information of the outcome, $x3 \sim \mathcal{N}(0, 1)$. The second, $x4$, is the outcome variable corrupted with random noise from a uniform distribution and therefore carrying some information about the outcome, $x4 = y + \epsilon, \epsilon \sim \mathcal{U}_{[0,1]}$.

Both the LSAM and compositional ensemble model are trained on the 4-dimensional dataset. Outcome measures are then bootstrapped with experiments repeated 30 times. We can obtain a statistical measure of distance between groups reported using a Student t-test and measure the $p$-value.

| $u_k$ | LSAM | | Ensemble | |
|---|---|---|---|---|
| | {} | $p$-value | {} | $p$-value |
| {x1} | 3.69 | $3.95 \times 10^{-10}$ | 0.29 | $4.43 \times 10^{-5}$ |
| {x2} | 4.18 | $2.16 \times 10^{-11}$ | 0.38 | $3.40 \times 10^{-4}$ |
| {x1, x2} | 5.10 | $4.13 \times 10^{-14}$ | 2.74 | $6.78 \times 10^{-5}$ |
| {x3 (noise)} | 3.44 | $1.14 \times 10^{-8}$ | 0.04 | $1.04 \times 10^{-3}$ |
| {x4 (signal)} | 5.15 | $3.77 \times 10^{-14}$ | 4.37 | $6.08 \times 10^{-5}$ |

Table 1: Euclidean distance between latent space representations of $u_k$ and the empty set for the LSAM and Ensemble based models. The $p$-value represents the distributional difference. The difference is greatest where the signal is strongest, and least where there is only random noise.

We first explore the Euclidean distance for various subsets compared to the empty set. From Propositions 1 and 2, we would predict that the distance is shorter when the subset is only random noise, $x3$, compared to carrying some information about the outcome. Table 1 shows that this is indeed the case, with the distance between the empty set and noise being the smallest value for both the ensemble and the LSAM. The statistical difference between the bootstrapped distributions is much less for the non-informative noise compared to subsets carrying information.

We can then explore the Euclidean distance between $u_k$, where $u_k \subseteq \{x1, x2\}$, and $u_k + x$, where $x \subset \{x3, x4\}$. From Propositions 1 and 2, we predict that the distance is shorter when $x$ is random noise, as in $x3$, compared to carrying information about the outcome, as in $x4$. This is confirmed in Table 2 where we see that adding noise changes the location of the mapping in latent space less than adding an informative variable across both models.

| $u_k$ | LSAM | | | Ensemble | | |
|---|---|---|---|---|---|---|
| | +{noise} | +{signal} | $p$-value | +{noise} | +{signal} | $p$-value |
| {x1} | 0.43 | 4.16 | $3.09 \times 10^{-12}$ | 0.16 | 2.41 | $8.24 \times 10^{-5}$ |
| {x2} | 0.37 | 3.18 | $1.02 \times 10^{-11}$ | 0.14 | 3.92 | $1.80 \times 10^{-4}$ |
| {x1, x2} | 0.21 | 1.09 | $3.77 \times 10^{-7}$ | 2.63 | 3.79 | $1.43 \times 10^{-2}$ |
| {} | 3.44 | 5.15 | $2.02 \times 10^{-7}$ | 0.04 | 4.37 | $6.28 \times 10^{-5}$ |

Table 2: Euclidean distance between latent space representations of $u_k$ and $u_k$ + either a noise variable or an informative variable for both a LSAM or Ensemble based model. The location in latent space moves the most when an informative variable is added to a subset.

### 6.1.2 MISSINGNESS REGULARIZATION

We now explore the effect of the level of missingness on the latent space representation. In order to do this we measure the Euclidean distance between $u_k$, where $u_k \subseteq \{x1, x2\}$, and $u_k + \{x4\}$, where the degree of missingness is increased in $x4$ from 0% of the data to 99% of the data. From Proposition 2, we would predict that as missingness in $x4$ increases, the distance between $u_k$ and $u_k + \{x4\}$ decreases. Table 3 empirically demonstrates this effect where the location in latent space converges to approximately the same location by 99% missingness.

### 6.1.3 MISSINGNESS, FEATURE IMPORTANCE AND CONCRETE DROPOUT

Finally, we explore the effect of missingness on the concrete distribution learned during training. Specifically, we can look at how the probability of dropping $x4$ from subsets during training changes as the level of missingness increases from 0% to 99% of the variable. Table 4 shows that the probability of dropout increases toward 0.5, the point of maximum entropy, as missingness increases to 99%. The implication of this finding is that the model assigns more importance to variables with less missing data, even when the variable with missing data is an informative variable.

| $u_k$ | 0% | 20% | 40% | 60% | 80% | 99% |
|---|---|---|---|---|---|---|
| {x1} | 7.84 | 5.93 | 4.62 | 2.88 | 1.48 | 0.07 |
| {x2} | 6.11 | 4.42 | 3.40 | 2.24 | 1.05 | 0.05 |
| {x1, x2} | 2.07 | 0.79 | 0.57 | 0.23 | 0.13 | 0.01 |
| {} | 9.13 | 6.88 | 5.42 | 3.45 | 1.77 | 0.09 |

Table 3: Euclidean distance between latent space representations of $u_k$ and $u_k$ + an informative variable ($x4$) for varying levels of missingness in the informative variable ($x4$). The location of the subset in latent space moves less when the missingness in the added informative variable increases.

| Variable | 0% | 20% | 40% | 60% | 80% | 99% |
|---|---|---|---|---|---|---|
| x1 | 0.34 | 0.33 | 0.32 | 0.33 | 0.33 | 0.34 |
| x2 | 0.32 | 0.32 | 0.32 | 0.32 | 0.31 | 0.33 |
| x3 (noise) | 0.50 | 0.49 | 0.49 | 0.49 | 0.48 | 0.49 |
| x4 (signal) | 0.40 | 0.38 | 0.37 | 0.38 | 0.40 | 0.46 |

Table 4: Learned probabilities, from a concrete distribution, of dropping variables during training across different levels of missingness in the informative variable ($x4$). The probability of dropping $x4$ from training approaches that of random noise ($x3$) as the missingness increases in that variable.

## 6.2 BENCHMARKING PERFORMANCE

The OpenML-CC18 classification benchmarking suite is used to assess performance in two settings (Vanschoren et al. 2013).

### 6.2.1 DATASET SELECTION

In the first set of experiments, we only select datasets with complete data availability, so we can controllably corrupt them with specific missingness patterns. In the second set of experiments, we select datasets with missingness, however the missingness pattern is unknown. For all experiments, we select tabular datasets with a maximum number of 250 features and rows between 1000 and 10000 for computability. For datasets with pre-existing missingness we included datasets with greater than or equal to 5% of the data missing. Our final collection includes 18 complete datasets and 11 datasets with missingness. Details on the methods used to corrupt the datasets with missingness are available in the supplementary materials.

### 6.2.2 COMPARATOR MODELS

In all experiments, we compare 'out-of-the-box' performance with missingness against industry standard data processing pipelines with imputation. We compare our LSAM approach against a high performing ensemble decision tree model, LightGBM, which is also capable of dealing with missing data natively (Ke et al. 2017).

Imputation can be broadly grouped into simple imputation strategies or regression methods. Simple imputation can be the replacement of a missing value by a statistic, a pre-specified value, or encoding the missingness as a new variable. Examples include mean or mode imputation. This strategy can lead to consistent predictions in the setting of MCAR, but is not appropriate where data is MAR or MNAR (Bertsimas et al. 2021). Despite the strong assumption needed, the benefit to this approach is computational efficiency and ease of implementation.

Regression strategies are often combined with multiple imputation techniques, which involves regressing multiple times with many models in order to generate a sample distribution of imputed values (Rubin 1987; Buuren and Groothuis-Oudshoorn 2011). Multiple imputation can be performed with linear regression, decision tree, or even random forest models and can provide unbiased estimates in the setting of data MCAR and MAR (Stekhoven and Buhlmann 2011). Although this approach can require significant computational resources, especially in the setting of large datasets, it has become the gold standard for most modeling tasks that require imputation. In our experiments, we compare

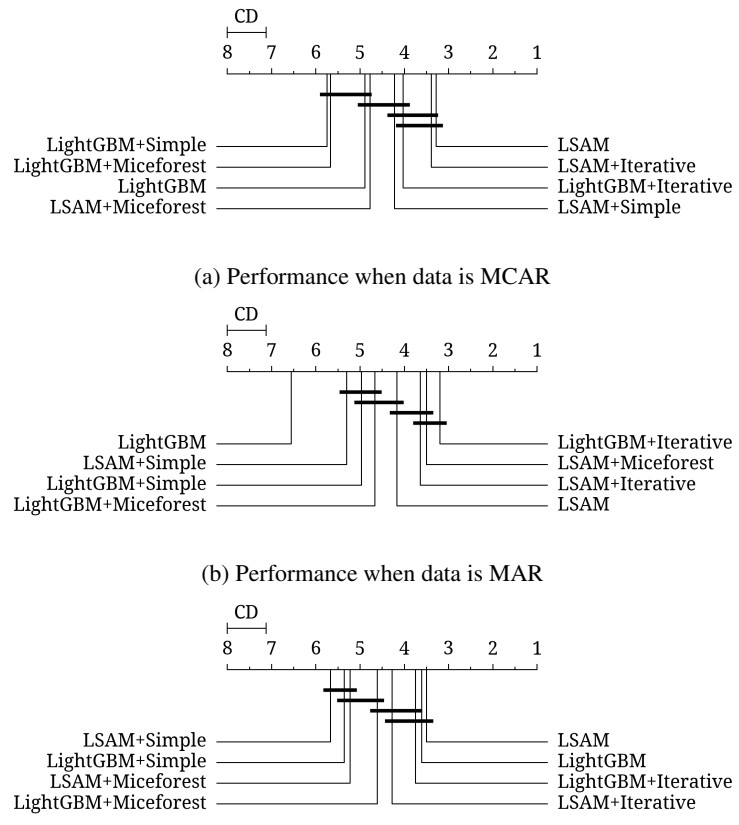

(a) Performance when data is MCAR

(b) Performance when data is MAR

(c) Performance when data is MNAR

Figure 1: Critical difference diagrams comparing performance for different missingness regimes, demonstrating improved performance for the LSAM without imputation. Points are labelled by the type of model as well as the imputation strategy if used. The performance metric is the change in negative log-likelihood from baseline performance with complete data. Further right in the diagram indicates better performance. A break in the solid bar underneath demonstrates statistical significance.

native model performance against 3 imputation strategies: simple imputation with mode or mean imputation based on data type, an iterative multivariate imputation strategy, and multiple imputation with random forests (MiceForest) (Pedregosa et al. 2011; Stekhoven and Buhlmann 2011).

### 6.2.3 PERFORMANCE ON COMPLETE AND CORRUPTED REAL-WORLD DATASETS

We first report the performance on the baseline complete datasets. For the metric of accuracy, the LSAM performs better with 13/18 wins. For the metric of negative log-likelihood (NLL) the LSAM is better with 12/18 wins.

To statistically compare the performance of all possible approaches we use critical difference diagrams with a Nemenyi two-tailed test (Demšar 2006). Figure 1 shows these plots in the context of the negative log-likelihood and demonstrates that the LSAM, using out-of-the-box performance, outperforms or performs as well as an 'impute and regress' strategy when data is MCAR or MNAR. Additionally, the LSAM outperforms LightGBM in most settings, with the exception of data MAR. Critical difference diagrams with the metric of accuracy are in the supplementary materials, and demonstrate similar findings.

### 6.2.4 PERFORMANCE ON REAL-WORLD DATASETS WITH MISSINGNESS

Again, we compare the results of all possible approaches using a critical difference diagram. Figure 2 shows these plots for both metrics of accuracy and negative log-likelihood. Performance as measured

by the negative log-likelihood showed a higher performance of the LSAM using out-of-the-box performance. Performance as measured by the accuracy showed a higher performance of LightGBM models. These results indicate that the LSAM model is better calibrated at the expense of accuracy.

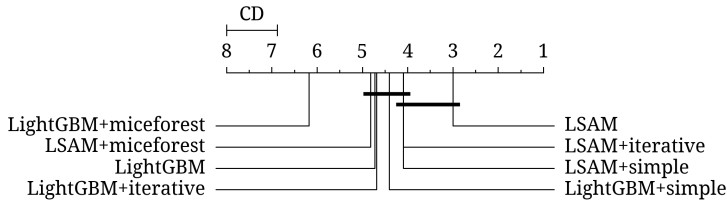

Figure 2: Critical difference diagram comparing performance with negative log-likelihood on datasets with unknown missingness pattern, demonstrating improved performance of the LSAM model.

### 6.2.5 META-LEARNING

Finally, to explore the effect of dataset characteristics when LSAM outperforms the comparators, we train a random forest model to predict an LSAM win from the known features of each dataset. Full experiment details and results are available in the supplementary materials. Datasets with higher dimensionality, a greater number of rows, and a greater proportion of numeric variables tended to favor LSAM.

## 7 CONCLUSION

To the best of our knowledge, our work is the first description and theoretical justification for utilizing an attention based latent space model for dealing with missingness in tabular datasets. We have shown that regularization occurs in the latent space both theoretically and empirically using a LSAM and compositional ensemble architecture. The regularization acts to cluster representations in the latent space toward the lowest cardinality and highest mutually informative subset of variables. We have shown this approach to outperform imputation and LightGBM.

Our results suggest that when data is MCAR and MAR, there is less of a clear advantage of avoiding imputation. Importantly, when data is MNAR, our experiments suggest a potential advantage of out-of-the-box methods to avoid the bias of imputation methods in this setting. This finding is predicted from our theoretical derivation of latent space regularization. Performance on unknown, and likely mixed, real-world missingness patterns corroborates the finding that latent space regularization can avoid the bias in imputation methods. Interestingly, LightGBM also outcompetes imputation when data is MNAR which likely arises from gradient based decision trees allocating a split with missingness to the direction that minimizes the loss (Ke et al. 2017). We suggest that as data MNAR contains some information about the underlying value it is likely that splitting based on loss minimization allows LightGBM to incorporate information from data MNAR.

A limitation of our approach is that it bundles missingness with model design. Certain predictive tasks with known missingness that is MAR or MCAR, or specific model requirements, would likely benefit from an impute and regress strategy instead of our proposed method. Additionally, while we have introduced latent space regularization to compositional ensemble and attention-based models, there is no limit to possible architectures that could achieve the same effect. Future work should further validate this latent space regularization with other model designs and explore the application of this approach in other real world dataset domains, such as imaging or time-series data.

We conclude that for predictive tasks, attention based latent space models with concrete dropout may be a principled model choice in the context of missing data with unknown missingness pattern. They require no assumptions about the missingness pattern of the data and require minimal extra computational cost compared to a similarly parameterized model.

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
