# OpenReview forum: "Dealing with missing data using attention and latent space regularization"
_ICLR.cc/2023/Conference — Submitted to ICLR 2023_

### Official Review · Reviewer_ezY7 · 2022-10-15

**Confidence:** 4
**Correctness:** 2
**Technical Novelty And Significance:** 3
**Empirical Novelty And Significance:** 3
**Recommendation:** 3

**Clarity, Quality, Novelty And Reproducibility:**

I found the paper hard to follow, as I could only slowly develop understanding of where the authors were trying to go, and after reading it fully, I still feel that I am second-guessing some of the assumptions and choices of the authors.

The manuscript starts heavy on the formalism, but light on the theoretical results. Where are the proof of prop 1 and 2?

With regards to novelty, I find that the paper has some novelty, but is not clearly positioning itself. Overall, it lacks clear formulations of both prior art and contribution.

The work should position itself with regards to the results of Le Morvan et al. "What’sa good imputation to predict with missing values?." NeurIPS 2021, as they are very relevant to the theoretical points made forward in the beginning of the paper.

With regards to reproducibility, the authors share the code, which should ensure reproducibility.


**Strength And Weaknesses:**


Modeling prediction in the presence of missing values, with suitable architectures is an important problem.

I feel that there are some implicit, or poorly-stated, assumptions behind the theoretical claims. For instance, it seems that I can create counter examples to prop 2: for a 5-dimensional feature space (X0, X1, X2, X3, X4), using Y = 1 is X0 is missing and 0 if not and f(U) = sum_U. I have then a non-zero mutual information I(U_i, Y) each time U_i contains X0, and elsewhere a zero mutual information, but this is quite independent of the value of f(U).

Reading the code source, I see 18 hyper-parameters for LSAM: regularization, optimization, and architectural hyperparameter (for f and g). How were they made for each experiment? In particular: what was the train-test-validation split strategy?

The fact that LSAM+iterative works as well as LSAM seems to me a bit contradictory with the spirit of LSAM, if I have understood things well. Indeed, I thought that the spirit of LSAM what to optimize jointly the representation of the missing values and the subsequent predictor.




**Summary Of The Paper:**

This submission studies an architecture to learn from data with missing values. The architecture is based on chaining two transformations with an attention module that enables modeling the set of observed values. The model learns by randomly masking the training data to ensure learning from different subsets of the variables.

The authors introduce information-theoretical considerations to justify the approach, reasonning on properties of embeddings in the presence of missing values.

The authors then empirically demonstrates their theoretical considerations and validate their approach on many datasets from openML with MCAR, MAR and MNAR simulated missingness by comparing the prediction accuracy to LightGBM.


**Summary Of The Review:**

The ideas of the papers are interesting, but they are not very well formulated, established, and positioned.

---

### Official Review · Reviewer_XuHA · 2022-10-24

**Confidence:** 2
**Correctness:** 2
**Technical Novelty And Significance:** 2
**Empirical Novelty And Significance:** 2
**Recommendation:** 5

**Clarity, Quality, Novelty And Reproducibility:**

As I have mentioned in the previous section, I think the clarity of the paper should be much improved for the general audience. Personally, I think the writing quality of this paper is not good.

I have trouble understanding the paper so I cannot safely judge the novelty. But according to algorithm 1, there is no significant novel part in the model formulation and training objectives. I guess the potential contribution is casting the missing value problem using measure theoretical language. But the author did not write it in a clear, digestible way.


**Strength And Weaknesses:**

Training models without imputation is an interesting point of view and using a measure theoretical point of view is novel to the best of my knowledge. But, I have to admit I am not sure I fully understand this paper. The clarity is a severe issue. Using measure theory to explain the missing mechanism should be fairly mathematical including the formal definition of the term used in the paper, claims made by the paper (or some references). However, most of the claims made by the paper are described in words, which is good for intuition but not good for formal explanation. Due to this, I got confused when I read this paper and still not clear on some of the claims made by the author.

For instance, in section 4, I am not sure why using latent space regularization helps remove the bias from missing mechanisms. Where are the proofs of propositions 1 and 2? I am not sure I understand the argument for countable additivity.

In section 5 paragraph 1, you mentioned defining a general function $f_\theta$ (since this is not has a subscript $k$, I assume it is shared for all ensembles.) but in the second paragraph, you define a feature-specific embedding network. Is the feature embedding network used to distinguish different ensembles? If so, these two are inconsistent. From algorithm 1, I assume $x$ contains some missing entries, how does the embedding network handle the missing part? In addition, why the transformer has to be trained on randomly generated subsets at each training step? Can this method be used for regression tasks, since the argument of mutual information is based on the cross-entropy loss function?

In section 2, you mentioned that without missingness, the measure is uniform. Just to double-check, the subset $u$ is the set containing variable indices (e.g. {1,3,5})?


**Summary Of The Paper:**

The author proposed a framework that treats model fitting with missing values as a latent space regularization problem using measure theoretical arguments. Specifically, the author translates the effect of missing values in the data space into the decreased mutual information between the latent variables and target variables. The author shows two propositions that mutual information is closely connected to the distances of latent variables in the latent space.
In the end, the author proposed an attention model as the encoder to map the observation into a latent variable with a feature embedding network.

Empirically, the author evaluates the proposed method for classification problems with a synthetic dataset and a benchmark dataset, showing improved results without imputation.

**Summary Of The Review:**

My main concern about this paper is the clarity and writing quality. Since I am not sure I fully understand the paper, I cannot judge the novelty or whether the contribution is significant.

---

### Official Review · Reviewer_4QuC · 2022-10-24

**Confidence:** 4
**Clarity, Quality, Novelty And Reproducibility:** See above.
**Correctness:** 3
**Technical Novelty And Significance:** 1
**Empirical Novelty And Significance:** 2
**Recommendation:** 3

**Strength And Weaknesses:**

Pros:

- This research follows the recent findings of supervised learning under missing data, and takes an approach that does not reply on imputation models
- I found that the information regularization perspective of latent space of missing data very interesting. I think additional works can be done based on this idea., especially on the theoretical side.

Cons:

- Clarity could be further improved. I understand that this paper focuses on supervised learning problem under missing data (i.e., prediction) as opposed to inference (i.e., multiple imputation) problems. However, this is not absolutely clear by just reading the title and abstract. The term "dealing with missing data" is too broad and confusing; and the presentation of the paper could further benefit from a more pronounced narrative.

- The conclusions/implications of the cited literature are not accurately presented. For example, regarding (Bertsimas et al. 2021), the authors claimed that "This strategy (simple mean imputation and then regress) can lead to consistent predictions in the setting of MCAR, but is not appropriate where data is MAR or MNAR" . This is very incorrect, as (Bertsimas et al. 2021) clearly proved and stated that "mean-imputation-then-regress is asymptotically consistent", and "the missingness mechanism (MAR or NMAR) does not impact asymptotic Bayes optimality". Although the authors could mention the limitation of such approach under finite data regime (which needs to support by empirical evaluations as well, which is lacking at the moment), the statement regarding consistency is clearly wrong.

- Lack of novelty and a lot of similar recent advantages in literature are ignored. For example, apart from impute then regress, there are also approaches that adopt the joint generative-discriminative approaches based on latent representations, such as Ipsen et al, 2022, Ma et al., 2019,  Ipsen et a., 2021 just to name a very few, which naturally satisfies latent regularization effect that this paper proposed. These approaches are also based on the so-called set function encodings of latent space that maps $U_k$ into $z$, which is quite comparable to the proposed transformer approach. The authors claimed that "our work is the first description and theoretical justification for utilizing an attention based latent space model for dealing with missingness in tabular datasets", which is not entirely true. For example, in Lewis et al 2021, transformers are already used to encode $U_k$ into $z$, and the learnt $z$ can be used for any downstream tasks such as prediction, imputation and acquisition, and can be trivially extended to fully supervised setting of this paper, as argued by Ipsen et al, 2022.

- Theoretical analysis. I appreciate that the authors provided analysis on the regularization/clustering effect of latent space, however this analysis is a little pale to justify the proposed algorithm, especially considering the highly related works mentioned above. This also make the connections between Section 4 and 5 a bit far-fetched. Further theoretical analysis on consistency and identification will be appreciated.

Other minor issues & questions.

- In algorithm 1, it appears that the model training requires access to fully observed (at least MCAR) data, which seems quite limited. Could the authors clarify?



References

- Josse, Julie, et al. "On the consistency of supervised learning with missing values." arXiv preprint arXiv:1902.06931 (2019).
- Ipsen, Niels, Pierre-Alexandre Mattei, and Jes Frellsen. "How to deal with missing data in supervised deep learning?." ICLR 2022
- Ma, Chao, et al. "Eddi: Efficient dynamic discovery of high-value information with partial vae." arXiv preprint arXiv:1809.11142 (2018).
- Ipsen, Niels Bruun, Pierre-Alexandre Mattei, and Jes Frellsen. "not-MIWAE: Deep generative modelling with missing not at random data." arXiv preprint arXiv:2006.12871 (2020).
- Lewis, Sarah, et al. "Accurate Imputation and Efficient Data Acquisitionwith Transformer-based VAEs." NeurIPS 2021 Workshop on Deep Generative Models and Downstream Applications. 2021.

**Summary Of The Paper:**

This paper studies the supervised learning problem under missing data, and proposed a solution without the need for imputation based by utilizing an attention based latent space model for dealing with missingness in tabular datasets. The authors shows that certain regularization occurs for the latent space representation of missing data with high cardinality subsets that are regularized to the higher information. Based on this insight, this paper then proposed a training algorithm based on transformers.

**Summary Of The Review:**

I believe this paper contains some interesting ideas that could be further explored. However, given its current state (misleading factual mistakes, lack of novelty) I am not convinced to recommend acceptance to the main conference.

---

### Official Review · Reviewer_cGGK · 2022-10-25

**Confidence:** 3
**Correctness:** 3
**Technical Novelty And Significance:** 3
**Empirical Novelty And Significance:** 2
**Recommendation:** 3

**Clarity, Quality, Novelty And Reproducibility:**

The paper has a reduced impact due to the lack of clarity. This makes difficult to identify the main novelty and quality of the technical contribution. Perhaps, those are the most important flaws that I see.

**Strength And Weaknesses:**

**Strengths:** The technical formulation of the problem as a measure-theory alike method is a valid point. I also liked the perspective where one identifies the bias induced by the missing samples in the latent space. There is also a reasonable amount of empirical results, however, the number of baselines considered is somehow small in my opinion.

**Weaknesses:** In general, I had a difficult time to identify the main contributions of the work, or at least to see what are the main differences with respect to previous ideas in the literature. The idea of fitting models that only use the information of observed variables under missing samples seems to not be correctly introduced wrt other previous works (i.e. MiWAE, PartialVAE or Eddi), or at least indicating what are the differences and what is new here. Additionally, the introduction of the measure theory formulation does not seem to provide a key advantage in the analysis, as in the end the authors use the mutual information between subsets for later using the attention method. From the theoretical aspect of missing data, I also feel that Le Morvan, 2021 should be cited as it is stated in the paper that there are no universally fair imputation methods (which somehow prevents the work from focusing in imputation). On the algorithmic part, it was also difficult to me to perceive what is special or at least, to understand the connection with the theoretical development.

Finally, I liked the design of the synthetic experiment, but the presentation of the results is not clear or sufficiently rigorous. I really would have preferred to see the metrics tables in the main manuscript, rather than all of them in the appendix. Details about the datasets used for benchmarks, number of samples and rates of missing data is also not clearly available.


MiWAE -- https://arxiv.org/pdf/1812.02633.pdf

PartialVAE -- http://bayesiandeeplearning.org/2018/papers/75.pdf

Eddi -- https://arxiv.org/pdf/1809.11142.pdf

Le Morvan, 2021 -- https://arxiv.org/pdf/2106.00311.pdf

**Summary Of The Paper:**

The papers introduces a framework for reasoning about model fitting and inference from observed variables when dealing with missing data. The main point seems to be the interpretation of the latent space representation using the mutual information between subsets which contain different elements of the dataset. A consequence of this interpretation is to intervene in the latent space using an attention-based representation. This makes the method to perform better than one baseline based on ensemble decision trees (LightGBM).

**Summary Of The Review:**

Good direction with interesting connections between missing data models and measure theory. In particular the use of the mutual information to identify potential effects in the latent space. However, there is a general noise around the contributions that make them less clear and difficult to assess. I recommend weak rejection, but I am open to discuss/understand better what are the main contributions, the differences wrt SOTA methods and related work.

---

### Official Review · Reviewer_Xkfr · 2022-10-26

**Confidence:** 4
**Correctness:** 2
**Technical Novelty And Significance:** 2
**Empirical Novelty And Significance:** 2
**Recommendation:** 3

**Clarity, Quality, Novelty And Reproducibility:**

Clarity (3/5): This paper is reasonably written, and a reader with related background should get the idea without significant issues. Some mathematical notations can be improved for better clarity. While formalising the problem with measure theory might be preferable to some readers, it introduces many convoluted notations (i.e. the random variable for missing dimension index), and most of these concepts are not linked and later used in the paper.

Quality (2/5): This paper is limited in its discussion and analysis of other solutions for the considered research problem and therefore doesn't provide a comprehensive justification for its contribution. Methods of dealing with missing values can trace to some early density-based approaches (http://mlg.eng.cam.ac.uk/zoubin/papers/nips93.pdf) to recent deep-based approaches (https://arxiv.org/abs/1807.03653). See (https://arxiv.org/pdf/2206.07769.pdf) for a more detailed list. The proposed method should at least be discussed with more related methods to indicate its advantages and application scenarios (at the moment, the proposed method is only motivated as a low-cost solution compared to the ensemble method). Also, the authors are unclear on whether the proposed method can adequately solve the conditions given in def 1 to def 3, while these definitions are provided and discussed at the beginning.

Novelty (2/5): This paper claims its novelty in the introduction of attention block in the proposed method, but it should be aware that similar ideas have been used before for missing data imputation (i.e. https://openreview.net/pdf?id=N_OwBEYTcKK). Furthermore, at the moment, the attention block seems to be a plug-in part after masking the previously learned feature from each dimension. I wonder if the author can indicate if attention usage here differs from other typical attention/transformer applications.

Reproducibility (4/5): The method should be quickly followed by the texts and algorithm 1, and experiments are based on public datasets. Therefore there should be a significant issue in reproducing most experiments. Public methods and experiment access would make things better.

**Strength And Weaknesses:**

(+) Making predictions with missing data is quite a challenging and undouble overlooked problem. This paper provides a working solution to address the problem with the attention mechanism.

(-) Regarding the missing data problem, this paper primarily focuses on the proposed method's framework. A systematic view of related work (e.g. density-based, vae-based, etc.) is not present in the discussion or the experiments. (See details below).

(-) While the proposed method acts as a working solution, this paper doesn't provide enough insights or justification of the method to either the problem or other candidate methods. The readers might not be convinced if the technique is the right choice (See details below).

**Summary Of The Paper:**

This paper considers the problem of training models capable of making predictions while there are missing values in input data.
The authors proposed a method that uses the attention mechanism as it can take an arbitrary number of features.
Experiments are conducted to investigate the learnt latent feature and the performance of the methods on various datasets.

**Summary Of The Review:**

Although the authors provide a working solution to the prediction problem with missing data, this paper doesn't give enough insight into the method. It cannot justify the method from existing approaches. So the paper is not considered as publishable as the readers might feel as unconvinced as the reviewer.

---

### Decision · Program_Chairs · 2023-01-20

**Decision:**

Reject

**Justification For Why Not Higher Score:**

This paper is a clear reject.

**Justification For Why Not Lower Score:**

n/a

**Metareview: Summary, Strengths And Weaknesses:**

This paper considers the problem of inference in the presence of missing data. The authors introduce a model for in which they interpret the latent space with measure- and information theoretic arguments. In experiments, they evaluate their proposed method.

The authors did not respond to the reviews and hence no discussion between reviewers and authors took place.

All reviewers agree that the considered problem setting is important and some also mention that the considered information theoretic perspective could be useful for further development of theoretical analysis. However, there are substantial concerns regarding the relation to existing methods - in the discussion as well as in the experiments. This renders the contributions unclear and justification of the proposed method is missing. Furthermore, there are some concerns about the theoretical results.

As the authors did not submit a rebuttal these concerns were not addressed. Hence, as all reviewers recommended rejection of the paper, I am also recommending its rejection.